# Short Communication: Taurine Long-Term Treatment Prevents the Development of Cardiac Hypertrophy, and Premature Death in Hereditary Cardiomyopathy of the Hamster Is Sex-Independent

**DOI:** 10.3390/nu16070946

**Published:** 2024-03-26

**Authors:** Ghassan Bkaily, Yanick Simon, Joe Abou Abdallah, Chaimaa Ouertane, Amina Essalhi, Abdelouahed Khalil, Danielle Jacques

**Affiliations:** 1Department of Immunology and Cell Biology, Faculty of Medicine and Health Sciences, Université de Sherbrooke, Sherbrooke, QC J1H 5N4, Canada; yanick.simon@usherbrooke.ca (Y.S.); joe.abou.abdallah@usherbrooke.ca (J.A.A.); chaimaa.ouertane@usherbrooke.ca (C.O.); amina.essalhi@usherbrooke.ca (A.E.); danielle.jacques@usherbrooke.ca (D.J.); 2Department of Medicine, Faculty of Medicine and Health Sciences, Université de Sherbrooke, Sherbrooke, QC J1H 5N4, Canada; abdelouahed.khalil@usherbrooke.ca

**Keywords:** taurine, cardiomyopathy, hereditary cardiomyopathy, hypertrophy, early death, heart failure, sex dependence

## Abstract

Recently, we reported that during the hypertrophic phase (230 days old) of hereditary cardiomyopathy of the hamster (HCMH), short-term treatment (20 days) with 250 mg/kg/day of taurine prevents the development of hypertrophy in males but not in females. However, the mortality rate in non-treated animals was higher in females than in males. To verify whether the sex-dependency effect of taurine is due to the difference in the disease’s progression, we treated the 230-day-old animals for a longer time period of 122 days. Our results showed that long-term treatment with low and high concentrations of taurine significantly prevents cardiac hypertrophy and early death in HCMH males (*p* < 0.0001 and *p* < 0.05, respectively) and females (*p* < 0.01 and *p* < 0.0001, respectively). Our results demonstrate that the reported sex dependency of short-term treatments with taurine is due to a higher degree of heart remodeling in females when compared to males and not to sex dependency. In addition, sex-dependency studies should consider the differences between the male and female progression of the disease. Thus, long-term taurine therapies are recommended to prevent remodeling and early death in hereditary cardiomyopathy.

## 1. Introduction

Throughout evolution, organisms have developed several defense mechanisms, such as antioxidants, to counterbalance and neutralize the harmful effects of reactive oxygen species (ROS). Maintaining the level of intracellular ROS within the physiological threshold is required for cell survival, redox homeostasis, and deficiency and could be implicated in the process of aging [1,2]. In general, the concept of a biological antioxidant refers to any compound that, when present at a concentration below that of an oxidizable substrate, can retard or prevent oxidation [3,4]. Indeed, antioxidants are molecules that can accept or donate electrons and hydrogen atoms to oxidants to stop their chain reactions [5,6]. They fall into two main groups: (i) secondary peroxide scavengers, derived from dietary intakes such as the nonessential sulfur amino acid taurine [7], and (ii) primary free radical scavengers, produced by the cell, such as superoxide dismutase, catalase, and, in particular, glutathione, as well as taurine [4,8,9].

There are two types of hypertrophies: physiological and pathological [10]. Hereditary cardiomyopathy is a progressive form of pathological hypertrophy associated with heart failure and premature death [11,12]. The animal model par excellence that mimics this pathology in humans is the UMX-7.1 cardiomyopathic hamster [11,12,13]. This model, which our group and others use, has been utilized extensively to understand cardiomyopathy better and find a treatment [11,12]. Several studies using this animal model of hereditary cardiomyopathy have been reported in the literature [11]. This pathology is associated with an increase in free radicals and a decrease in the natural antioxidant taurine. Although glutathione is the mother of endogenous intracellular antioxidants [14], taurine, a natural antioxidant and nonessential amino acid, sometimes called a conditional nonessential amino acid [14], is the most abundant among human body nutrients [8]. This is why it can be considered the guardian of redox balance. It is one of the few amino acids with a membrane transporter which also has a specific blocker [8,15]. This powerful antioxidant has been reported to block and prevent many pathologies related to the neural, renal, and cardiovascular systems [14,16,17]. Recently, using the UMX-7.1 cardiomyopathic hamster animal model, our group demonstrated that short-term taurine supplementation prevents the development of hypertrophy in males but not in females [18].

Furthermore, the degree of severity of hypertrophy and heart failure is higher in females than in males [18]. For these reasons, it is possible to postulate that the difference in the effect of short-term taurine treatment between males and females in hereditary cardiomyopathy is due to the extent of damage, which is more significant in females than in males, and the fact that long-term treatment may generate a sex-independent effect. In this study, we extended treatment from 20 (short-term) [18] to 122 (long-term) days. Our results show that long-term treatment with taurine prevents hypertrophy associated with heart failure and premature death independently of sex.

## 2. Materials and Methods

### 2.1. Animal Model, Study Design, and Taurine

The information concerning the material and methods used is similar to that reported recently [18]. In brief, taurine was purchased from Cedarlane Labs in Burlington, ON (Canada). All other chemicals were bought from Sigma Company in Oakville, ON (Canada).

Male and female UMX-7.1 cardiomyopathic hamsters (CM) are bred in the animal house of our Faculty of Medicine and Health Sciences at the University of Sherbrooke [18]. They are all treated with the same conditions (24 °C and 12/12 h: light/dark cycle) with access to a hamster diet and tap water. All experiments conform to the guidelines of the Canadian Council on Animal Care (Ottawa, ON, Canada) and are in accordance with our faculty ethical committee guidelines for using animals for research (approval date and code, respectively, 5 February 2024, 2023-2814). We used 233-day-old cardiomyopathic hamsters for a better comparison with our recently published work dealing with short-term treatment with taurine [18].

The animals were separated into four groups for the study of hypertrophy and four groups for the survival experiment, as described previously [18].

The 230-day-old HCMHs were treated for 122 days with tap water containing 125 mg/kg/day (or 12.5 mg/animal/day) or 250 mg/kg/day (or 25 mg/animal/day) of taurine. The taurine concentration was determined according to the recommended supplements [8]. The consumption of water/day was determined. Mortality was assessed daily and was weighed weekly.

### 2.2. Statistical Analyses

Means ± S.E.M. were compared using one-way ANOVA values, followed by the Bonferroni multiple comparison tests, where a *p*-value < 0.05 was considered significant. 

## 3. Results

### 3.1. Effect of 122 Days of Supplementation with Taurine (125 and 250 mg/kg/day or 12.5 mg/animal/day) on Heart Weight-to-Body Weight Ratio of Male and Female HCMHs

During the 122 days of taurine supplementation, we verified the water consumption in non-treated and taurine-treated male and female HCMHs. There were no significant differences in the consumption of water (mL/day) in the HCMHs in the presence or absence of taurine: non-treated male HCMHs: 27.3 ± 1.1 (*n* = 8); taurine-treated male HCMHs: 29.5 ± 1.2 (*n* = 3); non-treated female HCMHs: 30.8 ± 3.5 (*n* = 3); and taurine-treated female HCMHs: 34.5 ± 2.3 (*n* = 8). In the first protocol, we verified whether long-term treatment (122 days) with taurine supplementation blocks and prevents the development of cardiac hypertrophy in male and female HCMHs. Treatment with taurine for 122 days did not significantly affect the body weight of both the non-treated and treated male and female HCMHs (non-treated male HCMHs: 99.5 ± 9.1 g; non-treated female HCMHs: 95.2 ± 12.8 g; taurine-treated male HCMHs: 102 ± 11 g; taurine-treated female HCMHs: 86.9 ± 9.5). However, as seen in Figure 1 and as previously reported [18], in non-treated 233-day-old hereditary cardiomyopathic hamsters, the heart weight/body weight ratio increase was similar in males (7.09 ± 0.45, *n* = 9) and females (6.68 ± 0.21, *n* = 8). The treatment of male HCMHs for 122 days with taurine significantly (*p* < 0.0001) and ultimately prevented the development of cardiac hypertrophy (4.60 ± 0.18, *n* = 11). As in the treated males, in the female hereditary cardiomyopathy animals treated with taurine, we noted a significant (*p* < 0.01) decrease in the heart weight/body weight ratio (5.36 ± 0.11, *n* = 13). However, the decrease in the heart weight/body weight ratio induced by taurine was significantly (*p* < 0.01) more important in the males when compared to the treated females.

Using the same protocol described earlier, we tested the effect of long-term treatment with a high taurine concentration in the tap water (250 mg/kg/day or 25 mg/animal/day) as reported previously [18]. Our results were similar to those shown in Figure 1 for 125 mg/kg/day of taurine supplementation. The treatment of male HCMHs with 250 mg/kg/day taurine significantly (*p* < 0.0001) decreased their heart weight/body weight ratio (g/kg) from 7.35 ± 0.72 (*n* = 5) down to 3.94 ± 0.07 (*n* = 8). The treatment in females significantly (*p* < 0.05) decreased their heart weight/body weight ratio (g/kg) from 6.04 ± 20 (*n* = 8) down to 5.17 ± 0.24 (*n* = 4). The decrease was higher in males when compared to females (*p* < 0.0001).

### 3.2. Effect of Long-Term (122 Days) Treatment with Taurine (125 mg/kg/day or 12.5 mg/animal/day) on the Early Death of 233-Day-Old Male and Female HCMHs

Our recently published work showed that short-term (20-day) treatments with a high concentration of taurine (250 mg/kg/day or 25 mg/animal/day) prevented early death in both males and females [18]. In this series of experiments, we verified whether long-term treatment (122 days) with a taurine concentration (125 mg/kg/day) near the recommended supplemental dietary intake of taurine in humans (100 mg/kg/day) during the hypertrophic phase of hereditary cardiomyopathy [11,13] affected early death differently in male and female 233-day-old HCMHs. As Figure 2 shows, early death in the non-treated female HCMHs (*n* = 17) was significantly higher (*p* < 0.05) when compared to the male HCMHs (*n* = 16). During the 122 days of supplemental dietary taurine administration, there was a 49.5% rate of mortality in the non-treated male HCMHs (*n* = 13), compared to 90% in the non-treated females (*n* = 16). As can also be seen in this figure, treatment for 122 days with taurine significantly prevented early death in the male (*p* < 0.05) and female (*p* < 0.0001) HCMHs.

## 4. Discussion

Taurine is the most abundant nonessential amino acid in the body [8,16,17]. This nonessential amino acid is mainly present in many foods, particularly seafood, and in lesser amounts in meat [8,19,20]. Also, the heart and the brain produce taurine [21], and their contribution is insufficient; thus, supplemental dietary taurine is necessary even for a population that consumes seafood, such as the Japanese [8,22]. The internationally recommended taurine supplementation for healthy young men and women is 100 mg/kg/day [23]. However, there is no information on the concentration of supplemental dietary taurine in different diseases, mainly hereditary cardiomyopathy and early death. In addition, the internationally recommended intake of taurine supplementation in healthy humans is for six months or a maximum of one year. However, the duration of intake of taurine supplementation in pathological conditions such as heart failure and early death is not known. Our results showed that at the beginning of the development of hypertrophy (230-day-old animals), a taurine supplementation of 125 mg/kg/day (near the internationally recommended amount of 100 mg/kg/day) for 122 days of treatment prevented the development of cardiac hypertrophy in a sex-independent manner. Since the effects of low and high doses of taurine gave the same results, it suggests that the effect of taurine supplementation was independent of the concentration used. Thus, a higher concentration of 250 mg/kg/day is justified. This outcome is opposed to the effect of short-term supplemental dietary taurine, which we reported as being sex-dependent [18]. Thus, the absence of an impact of short-term supplemental dietary taurine on cardiac hypertrophy is not due to sex differences but to the higher rate of cardiac remodeling in female HCMHs when compared to male HCMHs. Although the heart weight/body weight ratio is similar in male and female HCMHs (Figure 1), the prevention of cardiac hypertrophy by taurine is higher in males than in females (Figure 1). We must mention that the body weight did not change between the male and female HCMHs, untreated or treated with taurine; however, the heart weight/body weight ratio significantly decreased in the taurine-treated animals compared to the non-treated animals. This indicates that the beneficial effect of long-term treatment with taurine is due to its prevention of cardiac hypertrophy. Furthermore, since the daily consumption of water with or without taurine was similar, it indicates that the consumption of taurine is the same, and the higher prevention of hypertrophy in male HCMHs compared to female HCMHs is not due to the higher consumption of taurine by male HCMHs compared to female HCMHs.

The process of the development of hypertrophy in hereditary cardiomyopathy could be at least in part different in male and female HCMHs. This difference, more particularly at the redox level, could be due to differences in the presence and density of NOXs [24] and the levels of sex hormones, which are known to possess an antioxidant effect [25,26]. Longer and early treatments with taurine may restore the heart to its normal status, which should be clarified in the future. Our results also showed that the impact of taurine on cardiac hypertrophy and early death is independent of the concentration of taurine used but dependent on the duration of supplemental dietary taurine. The duration of intake of this supplementation seems critical in females since long-term (this work) treatments, unlike short-term treatments [18], were more efficient in preventing cardiac hypertrophy in females. As seen in Figure 2, in the untreated animals, 90% of the female HCMHs died at 355 days of age; however, only 50% perished among males of the same age. These results indicate that, with respect to the early death of male and female HCMHs, the early mortality is higher in female than male HCMHs. These results showed that early death in HCMHs is sex-dependent. These results could be due to the level of cardiac damage in males and females [27,28,29].

We converted the dose given to the hamster to humans using the well-known dose conversion method [30]. Using this method, we determined that the equivalent dose for a human with a body weight of 70 kg is 17 mg/kg/day (125 mg/kg/day) and 33 mg/kg/day (250 mg/kg/day) of taurine. This dose is nearly 100 mg/kg/day, which is also recommended by the European Food Safety Authority (EFSA) [23]. In addition, the dose used in rats is near 100 mg/kg/day [31,32]. Although the equivalent dose for a human is lower than (but near) the one used in our experiments, any excess plasma taurine concentration will be eliminated by the kidney. We have to mention that the plasma taurine concentration is lower in 230-day-old HCMHs compared to normal 230-day-old hamsters, and supplementation with taurine restored the level of taurine to normal (unpublished results). Therefore, the beneficial effect of taurine is largely due to the normalization of its plasma level. More work should be carried out in order to determine the concentration of taurine in the cardiac tissues of untreated and treated HCMHs. 

## 5. Conclusions

Our work showed for the first time that it is highly important to distinguish between sex dependency and differences in the progression of the disease in males and females. In addition, our results also showed that long-term treatment with taurine is highly recommended to prevent and reverse the development of hypertrophy and early death in hereditary cardiomyopathy. Thus, our work showed that females should take supplemental dietary taurine for a longer time compared to males. However, long-term treatment with taurine is more efficient in preventing cardiac hypertrophy in females. Our results showed that the EFSA-recommended dose [23] could be used to treat hereditary cardiomyopathy in both men and women for at least one year. However, it is not known whether taurine supplementation should be continued beyond this. In addition, it is important to determine the functional and mechanistic pathways of taurine at the three phases of the disease (necrosis, hypertrophy, and heart failure associated with early death) in males and females. This should be verified in the future.

## Figures and Tables

**Figure 1 nutrients-16-00946-f001:**
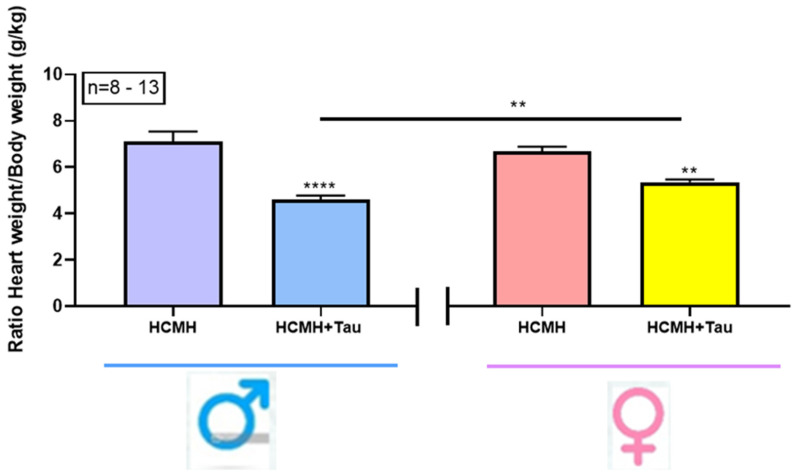
Effect of 122-day treatment of male and female HCMHs with 125 mg/kg/day of taurine on their heart weight/body weight ratio. Results expressed as mean ± S.E.M. ** *p* < 0.001 and **** *p* < 0.0001. Tau = taurine. *n* = number of animals.

**Figure 2 nutrients-16-00946-f002:**
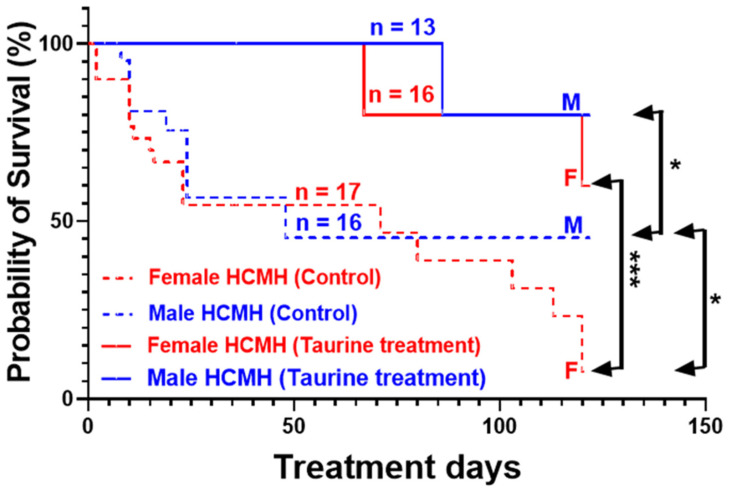
Effect of 122-day treatment (blue and red continuous lines) with 125 mg/kg/day of taurine dietary supplementation on the survival of male (M) and female (F) 233-day-old HCMHs. Values are expressed as percent survival, and *n* is the number of animals. * *p* < 0.05 and *** *p* < 0.001.

## Data Availability

The data presented in this study are available on request from the corresponding author.

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
