# Peer review of "Short Communication: Taurine Long-Term Treatment Prevents the Development of Cardiac Hypertrophy, and Premature Death in Hereditary Cardiomyopathy of the Hamster Is Sex-Independent"

_nutrients, 2024, doi:10.3390/nu16070946_

Round 1
Reviewer 1 Report
Comments and Suggestions for Authors
This study by Ghassan Bkaily demonstrates that taurine administration prolonged the survival of hamsters with cardiomyopathy. However, the authors need to further investigate and discuss the following points:
1. The currently recommended taurine dosage is specific to hamsters and has not been converted for humans. Since the pharmacokinetics of taurine varies significantly between different species, a comparison between hamsters and humans is necessary to determine the appropriate dosage for humans.
2. It is intriguing that taurine administration showed no effect in short-term treatment but prolonged survival in long-term treatment. This suggests that early administration may be effective in preventing the progression of early-stage cardiomyopathy to severe cases. The authors should discuss the differences between early and late-stage symptoms and the potential relationship between taurine's physiological effects (such as calcium homeostasis and ROS inhibition) and its therapeutic outcomes.
3. The authors should investigate the impact of different administration timings on cardiac function, comparing their findings with previous studies. This would help to clarify whether the timing of taurine administration, before or after the decline in cardiac function, affects its efficacy.
4. Detailed data on cardiac function and blood pressure are missing from the study. Analyzing these parameters could help to elucidate the mechanisms underlying the beneficial effects of taurine administration.
5. Measuring taurine concentrations in blood and myocardium would provide insights into its pharmacokinetics after administration.
6. Baseline data, such as body weight and food intake, of the animals before and after the experiment are not provided. This information is important for interpreting the effects of taurine administration.
7. In addition to heart weight, other heart failure markers, such as BNP and TnT, should be measured to evaluate the effects of taurine administration more comprehensively.
Author Response
REVIEWER 1
Response to Reviewer 1
Question 1: The currently recommended taurine dosage is specific to hamsters and has not been converted for humans. Since the pharmacokinetics of taurine varies significantly between different species, a comparison between hamsters and humans is necessary to determine the appropriate dosage for humans.
Answer 1: We thank the reviewer for his/her comments. As suggested, we converted the dose given to the hamster to humans using the well-known dose conversion method (Nair and Jacob in 2016). Using this method, we determined that the equivalent dose for a human with a body weight of 70 kg is 17 mg/kg/day (125 mg/kg/day) and 33 mg/kg/day (250 mg/kg/day) of taurine. This dose is nearly 100 mg/kg/day, which is also recommended by the European Food Safety Authority (EFSA). In addition, the dose used in rats is near 100 mg/kg/day (Ma et al. 2021; Silva et al. 2021). Although the equivalent dose for a human is lower (but near) of the one used in our experiments, any exceeding plasma taurine concentration will be eliminated by the kidney. This new information is added to the revised manuscript.
Question 2: It is intriguing that taurine administration showed no effect in short-term treatment but prolonged survival in long-term treatment. This suggests that early administration may be effective in preventing the progression of early-stage cardiomyopathy to severe cases. The authors should discuss the differences between early and late-stage symptoms and the potential relationship between taurine's physiological effects (such as calcium homeostasis and ROS inhibition) and its therapeutic outcomes.
Answer 2: As indicated in our manuscript, both the short (previously published paper)- and long (present manuscript)-term treatments started at around 230 days, which corresponds to the hypertrophy phase of the disease of the hamster (Bkaily and Jacques 2017). As suggested, we included this matter in the abstract and discussed it further in the revised manuscript.
Question 3: The authors should investigate the impact of different administration timings on cardiac function, comparing their findings with previous studies' findings. This would help to clarify whether the timing of taurine administration, before or after the decline in cardiac function, affects its efficacy.
Answer 3: We thank the reviewer for this suggestion. A week ago, we started a new trial (1.5-year duration) to explore what the reviewer is suggesting. We hope that this trial will help us clarify not only the effect of taurine on hypertrophy and heart failure but also at the early (necrosis phase) and the latter phase (heart failure) of the development of the disease. This will also permit us to explore taurine disease prevention's functional and mechanistic aspects.
Question 4: Detailed data on cardiac function and blood pressure are missing from the study. Analyzing these parameters could help to elucidate the mechanisms underlying the beneficial effects of taurine administration.
Answer 4: The reviewer is right that a functional study is needed, and this is clearly indicated in the conclusions section. Please see the answer to question 3. Our paper is a short communication, and we are allowed only for two figures with limited text.
Question 5: Measuring taurine concentrations in blood and myocardium would provide insights into its pharmacokinetics after administration.
Answer 5: We measured plasma taurine concentration before and at the end of treatment. Our results showed that the plasma concentration of taurine is very low compared to normal hamsters, and treatment with taurine brings the plasma concentration back to its normal level. This demonstrates that the correction of plasma taurine is the reason for its beneficial effect. This has been added to the discussion section since we cannot add a new figure to the short communication.
Question 6: Baseline data, such as body weight and food intake, of the animals before and after the experiment are not provided. This information is important for interpreting the effects of taurine administration.
Answer 6: We indicated the baseline data of the body weight. The consumption of water was similar in both treatment and non-treated animals. This information was added to the revised manuscript.
Question 7: In addition to heart weight, other heart failure markers, such as BNP and TnT, should be measured to evaluate the effects of taurine administration more comprehensively.
Answer 7: Our manuscript is a short communication, not a full paper. We limited our work to answer our hypothesis. We will for sure of all this in our ongoing trial (1.5 years). We did discuss this matter in our revised manuscript.
Reviewer 2 Report
Comments and Suggestions for Authors
In the present study, the authors sought to report the effect of a long term taurine treatment (122 days) in UMX-7.1 cardiomyopathic hamsters suggesting an amelioration the HCM progression and mortality rate in a sex-independent manner. The proposed topic is of pathophysiological interest. However, the novelty, other important points and limitation do not clearly emerge. Particularly:
Points
· As mentioned, the authors already reported through a short communication very similar findings. Indeed, the only variable indicated here regards the timing of the treatment, without providing more in-depth information, for example evaluating specific markers of HCM, or studying the anti-hypertrophic response (here only assessed by the heart weight/body weight ratio). On the other hand, here there is no the control group, as in the case of their first communication.
· The timing of the treatment should be justified and applied to human conditions. Indeed, 122 days in a hamster correspond to a much longer timescale than in humans. In this regard, the authors reported that in healthy humans the supplementation is carried out from 6 months to 1 year. How do the authors justify this important point? Are there any data that taurine supplementation can be carried out for longer periods in subjects with hereditary cardiomyopathy? If so, the authors should comment this aspect.
· Also the used doses should be supported by references. Furthermore, it is not clear whether the authors tested the dose of 250 mg/kg/day also for the long period. In this case this data should be included. The dose of 125 mg/kg/day needs to be justified by appropriate references and not considering a taurine dose near to the human supplementation of 100 mg/kg/day.
· Reference 7, reporting the taurine guidelines, is not sufficient to explain the doses and timing used in vivo. Please take care of this aspect.
· The current state, the work is too speculative, and results do not entirely justify the conclusions.
· Abstract should be reformulated.
· There are several typos in the text.
Comments on the Quality of English LanguageMinor editing of English language required.
Author Response
REVIEWER 2
We thank the reviewer for his/her helpful comments; we did revise the short communication according to the reviewer's comments.
Responses to Reviewer 2:
Question 1: The proposed topic is of pathophysiological interest. However, the novelty, other important points, and limitations do not clearly emerge.
Answer 1: As suggested, we made clear the novelty of our finding and the limitations.
Question 2: As mentioned, the authors have already reported very similar findings through a short communication. Indeed, the only variable indicated here regards the timing of the treatment, without providing more in-depth information, for example, evaluating specific markers of HCM or studying the anti-hypertrophic response (here only assessed by the heart weight/body weight ratio). On the other hand, there is no control group, as in the case of their first communication.
Answer 2: Our finding is completely the opposite of what we published in our previous short communication. The timing of treatment (nearly 223 days old) is the same in both previous short-term treatment and long-term treatment of this manuscript. The only variable between the two papers is the longer treatment time (122 days) in this manuscript compared to the short-term (20 days) in the previous paper. The objective of our short communication, as cited in the manuscript, is to test the hypothesis that the absence of taurine in female hypertrophy is not due to sex differences. It is due to the
fast development of hypertrophy in female cardiomyopathic hamsters (chronic state) compared to males (acute state). The control group is the non-treated cardiomyopathic hamsters.
Question 3: The timing of the treatment should be justified and applied to human conditions. Indeed, 122 days in a hamster corresponds to a much longer timescale than in humans. In this regard, the authors reported that the supplementation is carried out in healthy humans from 6 months to 1 year. How do the authors justify this important point? Are there any data that taurine supplementation can be carried out for longer periods in subjects with hereditary cardiomyopathy? If so, the authors should comment on this aspect.
Answer 3: As mentioned in answer 2, the timing of treatment is the same in males and females. As mentioned in our manuscript, we chose 230-day-old cardiomyopathic hamsters to study the development phase of hypertrophy similar to that of humans. This is now clearly cited. In healthy humans, taurine is used for athletic reasons. The idea of citing this study was to show that long-term treatment of healthy humans has no side effects. The is no data in the literature on hereditary cardiomyopathy or cardiomyopathy associated with early death at large dealing with taurine other than our previous and present short communications. There is no treatment for this disease, and our work, in the short and long term, will permit us to undergo a clinical trial which is already planned. All this information is taken into account in our revised manuscript.
Question 4: Also, the used doses should be supported by references. Furthermore, it is not clear whether the authors also tested the dose of 250 mg/kg/day for a long period. In this case, this data should be included. The 125 mg/kg/day dose needs to be justified by appropriate references and not considering a taurine dose near the human supplementation of 100 mg/kg/day.
Answer 4: As suggested, we converted the dose given to the hamster to humans using the well-known dose conversion method described by Nair and Jacob in 2016. ). Using this method, we determined that the equivalent dose for a human with a body weight of 70 kg is 17 mg/kg/day (125 mg/kg/day) and 33 mg/kg/day (250 mg/kg/day) of taurine. This dose is near 100 mg/kg/day, which is also recommended by the European Food Safety Authority (EFSA). This new information is added to the revised manuscript. The use of 250 mg/kg/day is known more clearly indicated in the revised manuscript. We added a new reference concerning 100 mg/kg/day used in the literature.
Question 5: Reference 7, reporting the taurine guidelines, is not sufficient to explain the doses and timing used in vivo. Please take care of this aspect.
Answer 5: Please refer to answer 4.
Question 6: Currently, the work is too speculative, and the results do not entirely justify the conclusions.
Answer 6: With all due respect to the reviewer, our work is not speculative, and we believe that our results justify (probably not entirely) our conclusions.
Question 7: Abstract should be reformulated
Answer 7: As suggested, we reformulate the abstract by taking into account the number of words allowed by the journal.
Question 8: There are several typos in the text.
Answer 8: We corrected the typos in the text using Grammarly.
Question 9: Minor editing of English language required.
Answer 9: The manuscript was revised by an expert in the English language.
Round 2
Reviewer 1 Report
Comments and Suggestions for Authors
This manuscript has been improved well.
Author Response
We are pleased that our revision was satisfactory to the reviewer.
Reviewer 2 Report
Comments and Suggestions for Authors
The Authors have responded to all point raised in R1, however information relating to the consumption of water/food intake and body weight should be added in the results and discussion section.
Author Response
We thank the reviewer again for his/her helpful comments. As suggested, we added the daily water consumption of non-treated and taurine-treated male and female HMCHs. There were no significant changes in water consumption. We also added the body weight of both untreated and taurine-treated male and female HCMHs. There were also no significant changes. All this information was added to the revised version of the manuscript and discussed in the discussion section. All the changes are highlighted in red.